

# Hypertension care cascade in the Ingwavuma rural community, uMkhanyakude District, KwaZulu-Natal province of South Africa

Herbert Chikafu and Moses Chimbari

School of Nursing and Public Health, University of KwaZulu-Natal, Durban, KwaZulu-Natal, South Africa

## ABSTRACT

**Background:** Treatment and control of hypertension are associated with a substantial reduction in adverse cardiovascular disease outcomes. Although South Africa aims to reduce the burden of cardiovascular diseases, there is limited evidence on the hypertension care cascade (HCC) performance in rural areas where stroke and hypertension are high. This study estimated HCC performance and identified predictors of hypertension screening among adults in the Ingwavuma community of KwaZulu-Natal, South Africa.

**Methods:** This was a cross-sectional study. Data were collected using the WHO STEPwise approach to surveillance (STEPS) questionnaire from 400 adult participants, excluding pregnant women and those with physical or cognitive impairments. Three hundred and ninety-three participants had complete data, and 131 had high blood pressure. We calculated progression rates for screening, diagnosis, treatment and control of hypertension from the sub-sample of participants with high blood pressure and assessed the bivariate association between HCC stages and participant characteristics and their effect sizes. We used binary and multivariable logistic regression to identify predictors of hypertension screening.

**Results:** Eighty-eight per cent of participants reported prior screening for hypertension. However, only 53.5% of patients under pharmacological treatment for hypertension had controlled blood pressure. In bivariate regression, employed participants were 80.3% (COR = 0.197, 95% CI [0.042–0.921]) more likely to be screened. In multivariable regression, the likelihood of hypertension screening was 82.4% (AOR = 0.176, 95% CI [0.047–0.655]) lower among participants in a cohabiting union than single participants. Similarly, employed participants were 87.4% (AOR = 0.129, 95% CI [0.017–0.952]) less likely to be screened than their unemployed counterparts.

**Conclusions:** The considerable attrition from the HCC across socio-demographic categories indicates a need for community-wide interventions. Empowering health care workers for community-based health promotion and hypertension management through point-of-care diagnostic tools could improve HCC performance. Efforts to improve the HCC should also focus on social determinants of health, notably gender and formal educational attainment.

Corresponding author
Herbert Chikafu,
chikafuh@gmail.com

# INTRODUCTION

The burden of disease in Sub-Saharan Africa and other low-to-middle income regions is shifting from communicable diseases to chronic noncommunicable diseases, and cardiovascular diseases cause substantial disability and premature death (*WHO, 2014*). Hypertension is most prevalent in Africa is a major risk factor for cardiovascular diseases (*WHO, 2019*). Poor outcomes for COVID-19 and for patients with hypertension, among other comorbidities, further highlight the importance of hypertension management. In a growing body of evidence, higher COVID-19 fatality was reported among patients with hypertension in a global systematic review (*Ssentongo et al., 2020*) and cohort study conducted in the Western Cape Province of South Africa (*Boulle et al., 2020*). Consequently, people with cardiovascular diseases and other chronic comorbidities have been listed among the prioritised groups in the COVID-19 vaccination program in South Africa (*Agency, 2021*).

Hypertension management improves cardiovascular disease outcomes (*Neal, MacMahon & Chapman, 2000*). However, management of hypertension is contingent on an effective hypertension care cascade (HCC) consisting of connected stages of screening, diagnosis, treatment and control. Care cascades for chronic conditions ensure timely intervention, consolidation of health gains and progression for further care where required (*Perlman, Jordan & Nash, 2016*). Despite the significance of continuum of care, studies conducted in sub-Saharan Africa reported significant attrition the out of the hypertension care cascade (*Gómez-Olivé et al., 2017*; *Ware et al., 2019*). Considerable spatial and socio-demographic variation in hypertension screening, awareness, treatment, and control in sub-Saharan Africa has been noted with poor outcomes in rural areas and among younger age groups (*Chow et al., 2013*; *Kayima et al., 2013*; *Lloyd-Sherlock et al., 2014*; *Price et al., 2018*). Progression through the HCC has been shown to be inversely associated with formal education, and females have a better awareness of their diagnosis, treatment, and control of hypertension than males (*Adeloye & Basquill, 2014*; *Chow et al., 2013*; *Gomez-Olive et al., 2013*; *Jardim et al., 2017*; *Kayima et al., 2013*). These care gaps in Africa have been attributed to health system deficiencies in most African countries (*Ibrahim & Damasceno, 2012*).

The suboptimal HCC performance in South Africa and other sub-Sahara African countries where improving life expectancy increases the population predisposed to metabolic-related diseases is worrisome given prevailing poor and pro-urban cardiovascular disease-related healthcare distribution (*Chikafu & Chimbari, 2019*; *Maredza et al., 2016*). Updated evidence on HCC performance is vital for timely health system interventions to avert avoidable fatal cardiovascular disease-related outcomes in rural areas with fewer or no tertiary health facilities. We, therefore, conducted this study in the Ingwavuma rural community, uMkhanyakude District of KwaZulu-Natal province, with the following three specific objectives: (a) to assess levels and gaps in the hypertension care cascade, (b) to assess

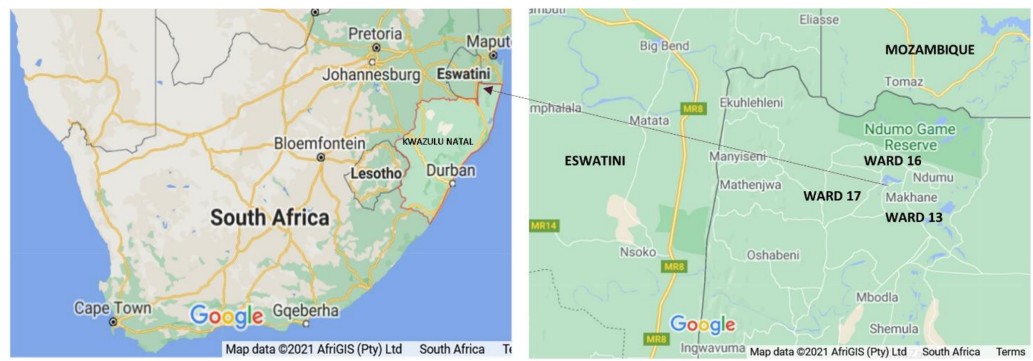

**Figure 1 Map of South Africa showing Ingwavuma rural community in north-eastern KwaZulu-Natal.** Map data © 2021 Google.

whether a relationship exists between socio-demographic characteristics and HCC outcomes, and (c) to determine predictors of hypertension screening.

## MATERIALS & METHODS

### Setting

We conducted this study in Ingwavuma, a rural community under Jozini Local Municipality in uMkhanyakude district Municipality of KwaZulu-Natal, South Africa. KwaZulu-Natal is the second-most populous province in South Africa and, notwithstanding inter-district variation, has the highest proportion of poor rural households and high unemployment levels (*Department of Statistics South Africa, 2019*; *Stats, 2018*). Although healthcare services are publicly provided, transport costs hinder the utilisation of healthcare services in rural KwaZulu-Natal (*Chimbindi et al., 2015*). UMkhanyakude district has a high prevalence of communicable and noncommunicable diseases (*Sharman & Bachmann, 2019*; *Wong et al., 2021*), and in mid-2021, it was among the districts with the least cumulative cases (11,548) and fatalities (422) of COVID-19 in KwaZulu-Natal province (*KZN_Department_of_Health, 2021*).

The Ingwavuma community is located along South Africa's border with Eswatini and Mozambique (Fig. 1), is semiarid and inhabited by IsiZulu speaking people under traditional leadership. Less than half of the study area population had attained secondary education, and most were unemployed (*Chikafu & Chimbari, 2020*). Formal healthcare services in rural Ingwavuma are largely publicly provided through 11 primary clinics within a 60 km radius of a referral district hospital that provides tertiary care. Patients access essential drugs freely at local clinics and monthly mobile clinics. Community health workers (CHW) trained in basic primary healthcare coordinate primary health activities in their communities. However, access to health centres is sometimes hindered by the long distances that the patients have to travel.

### Study design and participants

This was a cross-sectional, observational and analytical study. We determined a sample size of 384 participants through the Cochran formula for populations with unknown disease prevalence (*Israel, 1992*) and used a multistage sampling approach with a

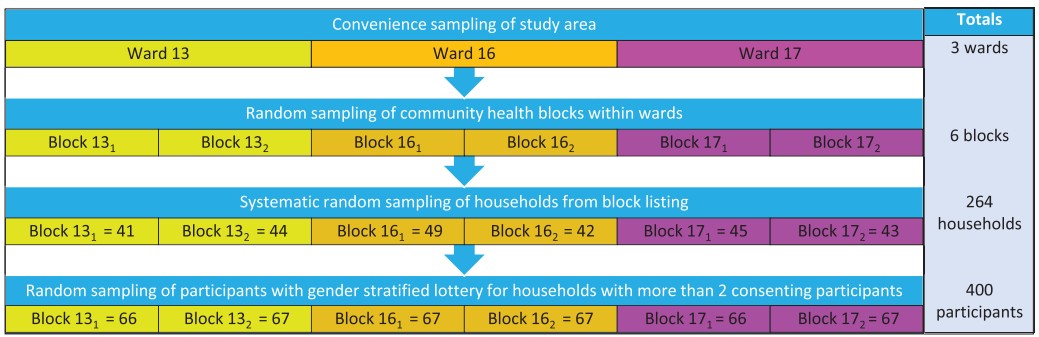

**Figure 2 Study sampling scheme.** Subscripts refer to randomly sampled household blocks in wards indicated by the index numbers.               

combination of convenience and random sampling (Fig. 2). Convenience sampling was used in the first stage to select the study area comprising three main villages that cover a considerable territory of the Ingwavuma community. The villages constitute three adjoining municipal wards (13, 16, and 17) of the Jozini local municipality. The decision to conduct the study in Ingwavuma was on the basis that the larger project in whose context the current study was conducted was situated in that area. In the second stage, two community health blocks were randomly chosen in each village. A community health block comprised of households served by a CHW. Households were then selected in stage three from a household listing of each block using systematic random sampling. After identifying a starting household picked from an online random number generator (*RANDOM.ORG, 2021*), we selected households with a sampling interval of one household. Households were selected only once, and a maximum of two adult participants (>= 18 years) enrolled from selected households after obtaining permission from the household were head or their proxy to enrol the household. We used the lottery method for households with more than two consenting participants with separate ballots for females and males. Only usual residents were considered in the study to ensure that findings were representative of the Ingwavuma community. The following groups were also excluded on ethical considerations or inability to attend measurement clinics: the physically and cognitively impaired, acutely ill and the aged. Pregnant women were not enrolled, considering the high likelihood of gestational induced hypertension and diabetes.

    A total of 400 adults enrolled in the main study, and 393 participants had complete data. We analysed hypertension cascade progression from 131 adults diagnosed with hypertension out of the 393 participants. The sub-sample of 131 adults shown in Table 1 comprised 89 (67.9%) females and 42 (32.1%) males with a mean age of 56.5 years (SD = 16.51). Over two fifths (42.7%) were at least 60 years old, a quarter (23.7%) had attained primary education, 55.7% reported to be single, and the majority (93.9%) were unemployed.

## Study instrument and data collection

We adapted the WHO STEPwise approach to surveillance (STEPS) questionnaire for this study (*WHO, 2010*). The questionnaire was translated into isiZulu by a native speaker,

**Table 1 Socio-demographic and clinical characteristics of adult participants diagnosed with hypertension in the Ingwavuma rural community (N = 131).**

| Variables and categories | | Female n (%) | Male n (%) | Total n (%) |
|---|---|---|---|---|
| **Overall** | | 89 (67.9) | 42 (32.1) | 131 (100) |
| **Age group** | 18–40 years | 18 (20.2) | 5 (11.9) | 23 (17.6) |
| | 40–59 years | 35 (39.3) | 17 (40.5) | 52 (39.7) |
| | >= 60 years | 36 (40.4) | 20 (47.6) | 56 (42.7) |
| **Education level** | None | 59 (66.3) | 19 (45.2) | 78 (59.5) |
| | Primary | 17 (19.1) | 14 (33.3) | 31 (23.7) |
| | Post-primary | 13 (14.6) | 9 (21.4) | 22 (16.8) |
| **Marital status** | Single | 56 (62.9) | 17 (40.5) | 73 (55.7) |
| | Married | 12 (13.5) | 9 (21.4) | 21 (16.0) |
| | Cohabiting | 21 (23.6) | 16 (38.1) | 37 (28.2) |
| **Occupational status** | Unemployed | 86 (96.4) | 37 (88.1) | 123 (93.9) |
| | Employed | 3 (3.4) | 5 (11.9) | 8 (6.1) |
| **Body mass index** | Up to normal weight | 27 (30.3) | 20 (47.6) | 47 (35.9) |
| | Overweight | 33 (37.1) | 14 (33.3) | 47 (35.9) |
| | Obese | 29 (32.6) | 8 (19.0) | 37 (28.2) |
| **Diabetes** | No | 51 (57.3) | 25 (59.5) | 76 (58.0) |
| | Yes | 38 (42.7) | 17 (40.5) | 55 (42.0) |
| **Physical activity level** | Insufficient | 17 (19.1) | 5 (11.9) | 22 (16.8) |
| | Sufficient | 19 (21.3) | 4 (9.5) | 23 (17.6) |
| | High | 53 (59.6) | 33 (78.6) | 86 (65.6) |
| **Alcohol consumption** | No | 77 (86.5) | 27 (64.3) | 104 (79.4) |
| | Yes | 12 (13.5) | 15 (35.7) | 27 (20.6) |
| **Tobacco use** | No | 87 (97.8) | 34 (81.0) | 121 (92.4) |
| | Yes | 2 (2.2) | 8 (19.0) | 10 (7.6) |

reviewed for appropriateness and pilot-tested in a village with community research assistants (CRAs) fluent in the local dialect. The finalised isiZulu questionnaire had closed and open-ended questions that were grouped into the following five sections; (a) socio-demographic characteristics, (b) health awareness, (c) health knowledge, (d) healthcare utilisation, and (e) modifiable health behaviours was pre-tested in one of the sampled villages. Households that participated in the pre-test were excluded from the study.

Data were collected between Novemeber 2020 and February 2021 over two phases. In the first phase, we conducted a household survey to collect self-reported data on health status and modifiable health behaviours (alcohol consumption, perception of salt consumption, and physical activity) using the STEPS questionnaire. In the second phase, blood pressure, blood glucose and anthropometric measurements were collected from phase one participants by a trained nurse at suitable locations within sampled communities. Data were collected electronically using KoBo collect on Android devices (*Initiative, 2018*) during both phases.

## Measurements and definitions

Brachial blood pressure was measured using a validated and calibrated digital blood pressure monitor (OMRON M6; Omron Healthcare Co, Japan) with appropriate cuff sizes. We recorded three diastolic and systolic pressure readings spaced at least 5 min after a short rest upon arrival at the measurement centre following WHO guidelines on the general environment and sitting posture (WHO, 2020). The second and third systolic and diastolic measurements were averaged to determine blood pressure status, and hypertension was defined as follows; (a) either or both of systolic blood pressure of ≥140 mmHg, diastolic blood pressure of ≥90 mmHg (25) or, (b) current use or history of pharmacological treatment prescribed by a healthcare professional to control hypertension.

We measured glycated haemoglobin A1c (HbA1c) from blood drawn from the second or third finger using a validated and calibrated point-of-care machine (HemoCue Hb 501, HemoCue, Angelholm, Sweden). The HbA1c test does not require fasting before an assessment and indicates mean blood glucose for a 2-to-3-month period preceding the test, and as such, it is ideal for screening undiagnosed patients and assessing blood sugar control for diagnosed patients. Diabetes mellitus status was classified as normal blood glucose (HbA1c; <5.7%), prediabetes (HbA1c; 5.7–6.4%) and diabetes (HbA1c; ≥6.5%) according to the WHO diagnostic criteria (WHO, 2016). Participants who self-reported current treatment for diabetes mellitus were classified as diabetic irrespective of HbA1c level.

The following anthropometric measurements were made; weight in kg to the nearest 100 g and height in cm using a calibrated digital scale (SECA 763; GmbH & Co. KG, Germany) fitted with a stadiometer. Participants wore light clothing, stood upright without shoes, facing horizontally forward and with feet and heels together, shoulders, head, and buttocks against the wall. Body mass index (BMI) was calculated and classified as underweight (BMI < 18 kg/m$^2$), normal weight (18 kg/m$^2$ ≤ BMI < 25 kg/m$^2$); overweight (25 kg/m$^2$ ≤ BMI < 30 kg/m$^2$); and obese (BMI ≥ 30 kg/m$^2$).

Participants also provided information on modifiable health behaviours. Physical activity (PA) level was determined from self-reported data on the type, intensity (moderate or vigorous) and duration (in hours and minutes) of physical activities during a typical week. Weekly estimates of physical activity were expressed in moderate-intensity minutes (1 vigorous-intensity minute = 2 moderate-intensity minutes) and categorised as insufficient (PA < 150 min), sufficient (150 ≤ PA < 300 min), and high (PA ≥ 300 min). Self-reported alcohol consumption and tobacco smoking in the month preceding the survey were used to define current alcohol consumption and tobacco smoking status.

## Outcome variables

The four stages of the HCC (see Fig. 3) were the outcome variables of the study. The care cascade depicts the progression in hypertension management. Participants were deemed screened if they ever had their blood pressure checked and classified as aware if they were diagnosed with hypertension and informed of their status by a health professional. Treatment was defined as the current use of antihypertensive medication prescribed by a

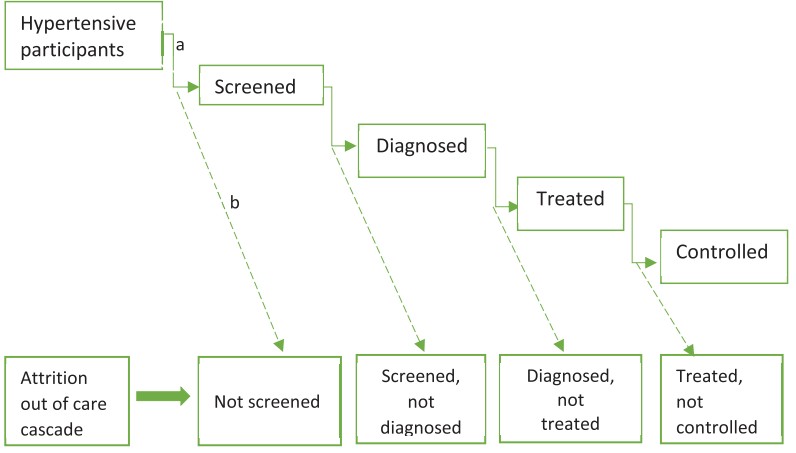

**Figure 3 The hypertension cascade of care.** Elbow arrow connectors (a) denote step-wise progression in the hypertension care cascade. Dashed arrows (b) depict step-wise attrition along the hypertension care cascade.

healthcare worker to manage blood pressure, while control was defined as systolic blood pressure <140 mmHg and diastolic blood pressure <90 mmHg on the testing day for participants under treatment.

## Statistical analysis

Data were analysed using IBM SPSS Statistics version 26 for Windows (IBM Corp., Armonk, NY, USA). Firstly, we assessed the prevalence levels of screening, awareness, treatment, and control. Hypertension care cascade performance was expressed in absolute and relative terms. Absolute progression rate was defined as the proportion of the 131 hypertensive participants achieving each level of the HCC while relative progression was expressed as the ratio of hypertensive participants attaining each level of the HCC relative to the preceding level. Secondly, the association between HCC stages and categorical factors was assessed using the chi-square test or Fisher's exact test, where applicable. We also obtained effect sizes to ascertain the strength of association and applied the Bonferroni and Holm-Bonferroni correction to correct for type 1 error risk. Thirdly, we conducted unadjusted and adjusted binary logistics regression analyses to determine the predictors of hypertension screening. Finally, we ran age-stratified logistic regressions to assess potential effect modifiers. Univariate regression results are presented as crude odds ratios (COR) and their 95% confidence intervals (95% CI), while multivariate regression results are presented as adjusted odds ratios (AOR) and 95% CIs. Statistical significance was set at $P < 0.05$ for all analyses.

## Ethical considerations

The study received ethical approval from the University of KwaZulu-Natal Biomedical Research Ethics Committee (BREC/00000235/2019), and participants provided written informed consent.

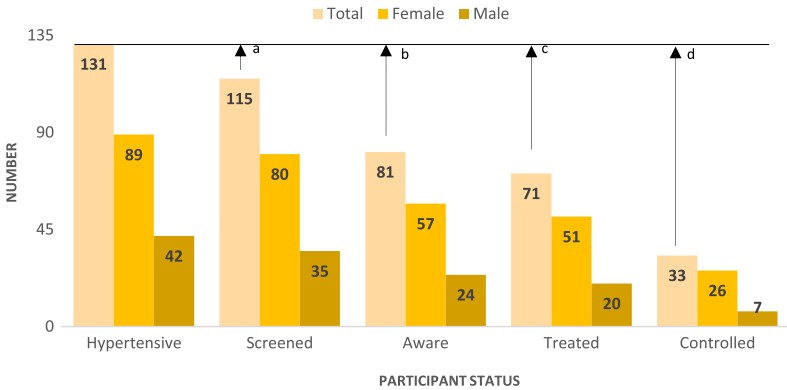

**Figure 4 Number of people at various hypertension care cascade stages out of a sample of 131 participants.** (a) Overall gap comprising of hypertensive participants who had not been screened. (b) Overall gap of hypertensive participants not aware of their hypertensive condition. (c) Overall gap of hypertensive participants not under pharmacological treatment for hypertension. (d) Overall gap of hypertensive participants with uncontrolled hypertension.

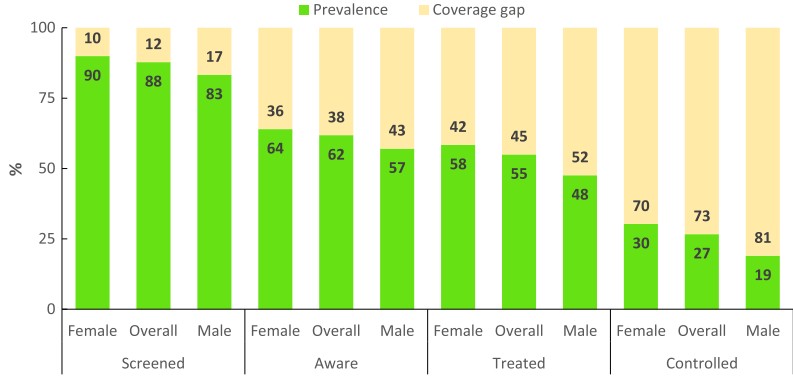

**Figure 5 Hypertension cascade of care showing progression rates and coverage gaps in the Ingwavuma community as proportions of the subsample (*N* = 131).**

## RESULTS

### Levels of screening, awareness, treatment, and control of hypertension

Attrition across the hypertension care cascade is noticeable in Ingwavuma. The number of participants progressing through the HCC levels are presented in fig. 4 and expressed as percentages of the cohort in fig. 5. Out of the 131 participants diagnosed with hypertension, 115 (88%) were screened, 81 (70%) were diagnosed, 71 (63%) were receiving treatment, and 33 (30%) had controlled blood pressure. In absolute HCC analysis, the least gap was observed between diagnosis and treatment, while the largest cascade gap was observed between treatment and control. However, relative progression rates were marginally higher, indicating treatment coverage of 87.7% among diagnosed participants and controlled blood pressure in half (53.5%) of patients receiving pharmacological treatment (Fig. 6).

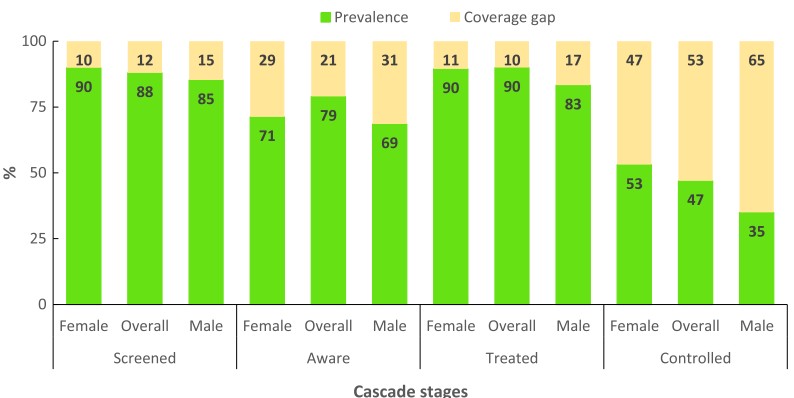

**Figure 6 Hypertension cascade of care showing stage-by-stage progression rates and coverage gaps in the Ingwavuma community.**

Table 2 presents HCC attainment levels across socio-demographic categories. The coverage of hypertension screening was at least 80% for all categories except for the 18–39 years (73.8%), post-primary education (72.7%) and employed (62.5%) categories. However, the prevalence of screening only differed significantly with employment status ($\chi^2 = 0.024$, d.f = 1), albeit with a small effect size (Cramer's V = 0.197). Further, there were comparable levels of awareness between females (71.3%) and males (68.6%); and an unexpectedly non-significant negative gradient between formal education and awareness. Awareness on hypertension status varied significantly with BMI ($\chi^2 = 0.006$, d.f = 2) albeit with small effect size (Cramer's V = 0.280). The Holm-Bonferroni correction ($P < 0.008$) indicated that overweight ($P = 0.005$) and obesity ($P = 0.002$) were associated with higher awareness. It is noteworthy that treatment coverage and control were not significantly associated with any predictor. However, control of blood pressure was higher among females (53.1%) than males (35.0%) and least (38.2%) among participants aged 60 years and above.

## Factors associated with screening for hypertension

Binary logistic regression showed that employment status was the only significant predictor of hypertension screening in the unadjusted analysis (Table 3). Compared with unemployed participants, being employed was associated with an 80.3% lower likelihood (COR = 0.197, 95% CI [0.042–0.921]) of being screened. In the adjusted regression model, employed participants remained less likely (AOR = 0.129, 95% CI [0.017–0.952]) to be screened relative to their unemployed counterparts. Although marital status was not a significant predictor of hypertension screening in the bivariate model, results showed that participants in a cohabiting union were 82.4% (AOR = 0.176, 95% CI [0.047–0.655]) less likely to have been screened compared with participants who reported to be single. In stratified regression to assess potential effect modifiers, marital status was a significant predictor among females only (Table 4). Females in a cohabiting union were 99% (AOR = 0.011, 95% CI [0.000–0.325]) less likely to be screened when compared with females who had never been in a marital union.

**Table 2 Levels and association of hypertension screening, awareness, treatment and control across socio-demographic factors in the Ingwavuma rural community.**

| Variable | | Screened n (%) | P value (ES)[a] | Diagnosed n (%) | P value (ES)[a] | Treated n (%) | P value | Controlled n (%) | P value |
|---|---|---|---|---|---|---|---|---|---|
| **Gender** | Female | 80 (89.9) | 0.285 | 57 (71.3) | 0.772 | 51 (89.5) | 0.443 | 26 (53.1) | 0.173 |
| | Male | 35 (85.3) | | 24 (68.6) | | 20 (83.3) | | 7 (35.0) | |
| **Age group** | 18–39 | 18 (73.8) | 0.186 | 10 (55.6) | 0.287 | 7 (70.0) | 0.095 | 3 (42.9) | 0.203 |
| | 40–59 | 45 (86.5) | | 34 (75.6) | | 29 (85.3) | | 17 (60.7) | |
| | 60+ years | 52 (92.9) | | 37 (71.2) | | 35 (94.6) | | 13 (38.2) | |
| **Formal education level** | None | 70 (89.7) | 0.053 | 55 (78.6) | 0.41 | 51 (92.7) | 0.128 | 22 (44.9) | 0.334 |
| | Primary | 29 (93.5) | | 18 (62.1) | | 14 (77.8) | | 9 (64.3) | |
| | Post-primary | 16 (72.7) | | 8 (50.0) | | 6 (75.0) | | 2 (33.3) | |
| **Marital status** | Not married | 67 (91.8) | 0.118 | 45 (67.2) | 0.136 | 38 (84.1) | 0.518 | 17 (45.9) | 0.733 |
| | Married | 19 (90.5) | | 17 (89.5) | | 15 (88.2) | | 8 (57.1) | |
| | Cohabiting | 29 (78.4) | | 19 (65.5) | | 18 (94.7) | | 8 (24.2) | |
| **Employment status** | Unemployed | 110 (89.4) | **0.024** (0.197) | 78 (70.9) | 0.631 | 69 (88.5) | 0.330 | 32 (47.8) | 0.950 |
| | Employed | 5 (62.5) | | 3 (60.0) | | 2 (66.7) | | 1 (50.0) | |
| **Diabetes mellitus** | No | 66 (88.8) | 0.698 | 50 (75.8) | 0.147 | 44 (88.0) | 0.904 | 22 (50.0) | 0.632 |
| | Yes | 49 (89.1) | | 31 (63.3) | | 27 (87.1) | | 11 (44.0) | |
| **Body mass index** | Normal weight | 41 (87.2) | 0.954 | 29 (70.7) | **0.002** (0.326) | 26 (89.6) | 0.621 | 13 (50.0) | 0.903 |
| | Overweight | 41 (87.2) | | 22 (53.6)[b] | | 18 (81.8) | | 8 (50.0) | |
| | Obese | 33 (89.1) | | 30 (90.9)[b] | | 27 (90.0) | | 12 (44.4) | |
| **Physical activity level** | Low | 18 (81.8) | 0.353 | 16 (88.9) | **0.06** | 13 (81.3) | 0.658 | 5 (38.5) | 0.755 |
| | Sufficient | 22 (95.7) | | 12 (54.5) | | 11 (91.7) | | 5 (50.0) | |
| | High | 75 (87.2) | | 53 (70.7) | | 47 (88.7) | | 23 (50.0) | |
| **Alcohol consumption** | No | 93 (89.4) | 0.261 | 66 (71.0) | 0.797 | 60 (90.9) | 0.062 | 27 (46.6) | 0.627 |
| | Yes | 22 (81.5) | | 15 (68.2) | | 11 (73.3) | | 6 (54.5) | |
| **Tobacco use** | No | 107 (88.4) | 0.434 | 75 (70.1) | 0.769 | 66 (88.0) | 0.738 | 32 (50.0) | 0.359 |
| | Yes | 8 (80.0) | | 6 (75.0) | | 5 (83.3) | | 1 (20.0) | |

**Note:**
[a] Cramer's V effect size for significant variables, [b] Statistically significant categories after Holm and Holm-Bonferonni corrections. Bold P-values indicate significant statistically variable.

# DISCUSSION

This study assessed the hypertension care cascade in the Ingwavuma rural community in north-eastern KwaZulu-Natal, South Africa. An effective hypertension care cascade is essential for early detection and management of hypertension, a major cardiovascular disease risk factor. In the backdrop of the high burden of stroke in rural South Africa (*Maredza et al., 2016*), efficient hypertension management could reduce the financial burden on households and the public healthcare system by minimising the often lengthy and costly tertiary care associated with cardiovascular disease complications.

Our study noted attrition at all stages of the HCC. The gaps in diagnosis, treatment and control are consistent with studies conducted in rural South Africa (*Addo, Smeeth & Leon, 2007*; *Adeniyi et al., 2016*; *Siedner et al., 2018*; *Wong et al., 2021*). Hypertension care cascade gaps in South Africa have been attributed to long distances to health centres, poor

**Table 3 Unadjusted and adjusted logistic regression analysis on hypertension screening prevalence among adults in the Ingwavuma rural community.**

| Variable | | COR[1] (95% CI) | P value | AOR[2] (95% CI) | P Value |
|---|---|---|---|---|---|
| **Gender** | Female | Reference | | | |
| | Male | 0.563 [0.194–1.631] | 0.289 | 1.272 [0.242–6.698] | 0.776 |
| **Age group** | 18–39 | Reference | | | |
| | 40–59 | 0.277 [0.067–1.145] | 0.076 | 0.199 [0.199–9.414] | 0.750 |
| | 60+ years | 0.495 [0.136–1.799] | 0.285 | 0.493 [0.493–32.302] | 0.194 |
| **Formal education level** | None | Reference | | | |
| | Primary | 1.657 [0.332–8.280] | 0.538 | 1.288 [0.183–9.039] | 0.799 |
| | Post-primary | 0.305 [0.093–1.001] | 0.050 | 0.399 [0.060–2.630] | 0.339 |
| **Marital status** | Not married | Reference | | | |
| | Married | 0.851 [0.159–4.562] | 0.850 | 0.670 [0.079–5.688] | 0.714 |
| | Cohabiting | 0.325 [0.103–1.020] | 0.054 | 0.126 [0.025–0.629] | **0.011** |
| **Employment** | Unemployed | Reference | | | |
| | Employed | 0.197 [0.042–0.921] | **0.039** | 0.129 [0.017–0.952] | **0.045** |
| **Diabetes mellitus** | No | Reference | | | |
| | Yes | 1.237 [0.421–3.635] | 0.698 | 1.907 [0.439–8.280] | 0.389 |
| **Body mass index** | Up to normal weight | Reference | | | |
| | Overweight | 1.000 [0.298–3.359] | 1.000 | 1.390 [0.231–8.368] | 0.719 |
| | Obese | 1.207 [0.314–4.637] | 0.784 | 2.665 [0.437–16.239] | 0.288 |
| **Physical activity level** | Low | Reference | | | |
| | Sufficient | 4.889 (0.501–47.708) | 0.172 | 8.173 [0.576–115.941] | 0.121 |
| | High | 1.515 [0.432–5.313] | 0.516 | 2.589 [0.526–12.752] | 0.242 |
| **Alcohol consumption** | No | Reference | | | |
| | Yes | 0.520 [0.164–1.651] | 0.268 | 0.284 [0.061–1.318] | 0.108 |
| **Tobacco smoking** | No | Reference | | | |
| | Yes | 0.523 [0.101–2.716] | 0.441 | 1.023 [0.120–8.750] | 0.983 |

**Note:**
[1]COR, Crude odds ratio; [2]AOR, Adjusted odds ratio; Bold P-values indicate significant statistically variables.

health-seeking behaviour and perceived low quality of services, including lengthy waiting periods (*Nulu, Aronow & Frishman, 2016*). Lifestyle interventions including weight reduction, healthy diet, physical activity and reduced alcohol consumption are efficacious for managing metabolic syndrome and accordingly, hypertension management guidelines in South Africa recommend lifestyle modification as the sole therapy for grade 1 hypertension (systolic BP; 140–159 mmHg and diastolic BP; 90–99 mmHg) for up to 6 months and as a complementary therapy for higher grades of hypertension (*Seedat, Rayner & Veriava, 2014*). Thus, the gap between diagnosis and treatment may not be an exclusive indication of HCC failure, but could also result from participants with elevated blood pressure under non-pharmacological treatment.

Although relatively higher than levels found in other rural areas in national studies (*Berry et al., 2017*; *Ware et al., 2019*), the low level of controlled hypertension in Ingwavuma is a concern. In South Africa, where most households depend on publicly

**Table 4 Gender stratified adjusted logistic regression analysis on hypertension screening prevalence among adults in the Ingwavuma rural community.**

| Variable | | P Value | |
|---|---|---|---|
| | | Female | Male |
| **Age group** | 18–39 | | |
| | 40–59 | 0.076 | 0.750 |
| | 60+ years | 0.285 | 0.194 |
| **Formal education level** | None | | |
| | Primary | 0.538 | 0.799 |
| | Post-primary | 0.050 | 0.339 |
| **Marital status** | Not married | | |
| | Married | 0.850 | 0.714 |
| | Cohabiting | **0.011** | 0.054 |
| **Employment** | Unemployed | | |
| | Employed | **0.039** | **0.045** |
| **Diabetes mellitus** | No | | |
| | Yes | 0.698 | 0.389 |
| **Body mass index** | Up to normal weight | | |
| | Overweight | 1.000 | 0.719 |
| | Obese | 0.784 | 0.288 |
| **Physical activity level** | Low | | |
| | Sufficient | 0.172 | 0.121 |
| | High | 0.516 | 0.242 |
| **Alcohol consumption** | No | | |
| | Yes | 0.268 | 0.108 |
| **Tobacco smoking** | No | | |
| | Yes | 0.441 | 0.983 |

Note:
Bold *P*-values indicate significant statistically variables.

provided healthcare, it is crucial to improve the level of controlled hypertension to minimise avoidable expenditure to enhance the country's expenditure efficiency of the constrained public health financing and improve the poor HCC outcomes relative to national income (*Geldsetzer et al., 2019*).

Social determinants of health explain most health differentials in South Africa (*Ataguba, Day & McIntyre, 2015*). Health awareness and outcomes are poor in communities with low socioeconomic status, particularly in rural and poor communities where formal education attainment is low. Contrary to earlier studies conducted in South Africa and other African countries (*Adeniyi et al., 2016*; *Chow et al., 2013*; *Siedner et al., 2018*), our study did not find notable differences between unadjusted and adjusted analyses on most socio-demographic predictors of hypertension screening except gender. The non-significance of most social determinants of health suggest the need for community-wide interventions to promote overall health-seeking and HCC progression. There is also a need for complementary structural measures to address

differentials in HCC progression. Low healthcare utilisation rates for noncommunicable diseases in rural areas has been attributed to the predominance of primary healthcare services concentrated on maternal health, child health and communicable disease management (*Geldsetzer et al., 2019*). Because of the chronic noncommunicable disease burden, it is necessary to promote primary healthcare beyond maternal health, child health and communicable disease control. Furthermore, health promotion interventions could be used to influence health perceptions and awareness to shift the utilisation of chronic healthcare services, including hypertension management from reactive utilisation toward proactive health-seeking.

Interventions to improve HCC outcomes should also focus on the social determinants of health to influence health-seeking behaviours, particularly gender. The better HCC performance among females observed in overall and sex-stratified analysis corroborates findings from national-level surveys (*Lloyd-Sherlock et al., 2014*; *Peltzer & Phaswana-Mafuya, 2013*; *Steyn et al., 2008*) and is consistent with findings from studies conducted in rural municipalities in northeastern South Africa (*Gomez-Olive et al., 2013*; *Jardim et al., 2017*) and other sub-Saharan African countries (*Gómez-Olivé et al., 2017*; *Kayima et al., 2013*; *Price et al., 2018*). Adult females have higher odds of better HCC performance than males partly resulting from frequent contact with the healthcare system for maternal healthcare needs. The gap between males and females is also attributable to gendered household roles in rural areas where females attend to family health needs. Often, women accompany young and elderly patients to seek care and thus have more contact with the healthcare system. Therefore, all else remaining the same, women have a higher likelihood of being screened for hypertension and progress through the hypertension care cascade. On the other hand, cultural connotations of masculinity and power that were found to affect general health-seeking and HIV treatment among males in rural KwaZulu-Natal (*Hunter, 2003*) potentially explain their poor performance on the HCC. Therefore, it is crucial to embed strategies to reshape gendered health-seeking perceptions among males.

Implementation of primary healthcare reforms has the potential to improve HCC outcomes in Ingwavuma and other rural areas. The reach and scope of CHWs evolved considerably over the past century from limited roles as "Native Anti-Malaria Assistants" in selected regions (*MacKinnon, 2001*) to the current Ward-based Primary Health Care Outreach Team (WBPHCOT) Strategy where they provide the last-mile delivery of primary health, including promoting treatment adherence for chronic conditions (*Health NDo, 2018*; *Schneider et al., 2018*). In addition to executing treatment adherence strategies towards attaining the WBPHCOT policy outcome of longevity and good health (*Health NDo, 2018*), there is a need to incorporate health promotion strategies to improve awareness of hypertension and its nexus with noncommunicable diseases in rural communities. Regional health promotion units could also provide CHWs with intelligible synthesised health system reports and research findings for appraisal with national and regional epidemiology patterns. Furthermore, local government and public health administrators should encourage researchers to undertake community engagement activities.

Advances in diagnostic technology provide opportunities to improve hypertension care cascade performance in rural areas. Given widespread white coat and masked hypertension in Africa (*Noubiap et al., 2018*), portable blood pressure monitors could be used for periodic home-based blood pressure screening and monitoring. Such community-based initiatives could help manage overcrowding at health facilities and allow physical distancing in line with COVID-19 protocols. They also improve access to care in areas where primary healthcare centres are distant. Similarly, given that mobile and internet connectivity is relatively good in South Africa, including our study area, SMS-text messaging and other mHealth innovations could be used to provide support and improve treatment adherence. A study conducted in South Africa demonstrated a positive effect of SMS-text messaging on hypertension treatment adherence (*Leon et al., 2015*). Furthermore, the community-based and eHealth initiatives could be integrated with geographic information systems to generate surveillance data for policymaking.

This study had some limitations. First, we used self-reported data that may have misestimated the prevalence of screening and diagnosis due to recall bias. Second, the study was cross-sectional, and as such, we could not infer causality. Third, we determined hypertension status from a single sitting due to resource constraints against the recommended three distinct moments in standard clinical diagnosis (*Seedat, Rayner & Veriava, 2014*). However, the study had noteworthy strengths. The major strength of this study is the generation of micro-level evidence on HCC performance from a large sample in a rural community characterised by low socioeconomic indicators and dominant traditional practices. Micro-level patterns in healthcare utilisation can be missed in large scale surveys. Also, excluding temporary residents from the study ensured that findings reflect HCC levels in the community. Participants who affirmed being on prescribed antihypertensive treatment during the household survey were requested to avail their medication packets for verification, and as such, our study provides a reliable estimate of hypertension control in the Ingwavuma community. Finally, the study site has socio-demographic similarities with other rural municipalities in the uMkhanyakude district municipality thus, our study findings are generalisable to the district level.

## CONCLUSIONS

We conclude that there is a sub-optimal hypertension cascade of care in Ingwavuma, especially the low level of controlled hypertension for patients under treatment. The findings have implications on the government's commitment towards reducing the burden of noncommunicable diseases, achievement of sustainable development goals and targets for noncommunicable diseases in the National Development Plan. Furthermore, improvements in the HCC are necessary for managing the COVID-19 pandemic. We recommend the following to improve the hypertension care cascade in rural KwaZulu-Natal where supply-side constraints and socioeconomic factors limit utilisation of healthcare services: (a) skilling of community health workers to promote awareness on hypertension and chronic noncommunicable diseases, (b) incorporation of hypertension management programs into the existing clinic and community-based health promotion activities, (c) adoption of point of care technology for community-based

screening and monitoring of blood pressure, and (d) adaption of eHealth and mHealth interventions to promote health awareness and treatment adherence.

## ACKNOWLEDGEMENTS

The authors thank the Ingwavuma community leaders and households for participating in this research. We are also grateful to Celiwe Tembe, Nozipho Mthembu, and Sinenhlanhla Mthembu for their meticulous work as Community Research Assistants. We acknowledge, with gratitude, TIBASA colleagues for their support.

### Funding

This research was funded by Tackling Infections To Benefit Africa (TIBA) through grant UoERef:CT–4987(d) and the University of KwaZulu-Natal through a PhD studentship bursary awarded to Herbert Chikafu by the College of Health Sciences. The funders had no role in study design, data collection and analysis, decision to publish, or preparation of the manuscript.

### Grant Disclosures

The following grant information was disclosed by the authors:
Tackling Infections To Benefit Africa (TIBA): UoERef:CT–4987(d).
University of KwaZulu-Natal, College of Health Sciences.

### Competing Interests

The authors declare that they have no competing interests.

### Author Contributions

- Herbert Chikafu conceived and designed the experiments, performed the experiments, analyzed the data, prepared figures and/or tables, authored or reviewed drafts of the paper, and approved the final draft.
- Moses Chimbari conceived and designed the experiments, authored or reviewed drafts of the paper, supervised the study, and approved the final draft.

### Human Ethics

The following information was supplied relating to ethical approvals (*i.e.*, approving body and any reference numbers):

This study was approved by the University of KwaZulu-Natal Biomedical Research Ethics Committee (BREC) (BREC/00000235/2019).

### Data Availability

The raw data are available as a Supplemental File.

## Supplemental Information

Supplemental information for this article can be found online at http://dx.doi.org/10.7717/peerj.12372#supplemental-information.

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
