# Peer review of "Hypertension care cascade in the Ingwavuma rural community, uMkhanyakude District, KwaZulu-Natal province of South Africa"

_PeerJ, doi:10.7717/peerj.12372_

## Round 0.1 · original submission · Major Revisions

Reviewers have considered your work. They found a lot of merits but indicated concerns that have to be addressed. Please, authors, provide detailed feedback to address all concerns. Looking forward to your revised manuscript. Thank you very much

·

Basic reporting

1. There is no flow/continuity in the introduction and hence it can be reframed.

Instead, the reframed of the introduction in the following way can be considered:
- Globally and the country wise burden with current Covid 19 situation.
- What is HCC? Importance of HCC and its effects.
- Influencers of HCC, screening and control.
- The Need to assess / the rationale followed by objectives.

Experimental design

1. Study Design:
Before writing about study instrument the author could have mentioned ,
• Study design
• Sampling frame
• Study participants
• Inclusion & exclusion criteria
• Study period
• Sample size
2. Settings:
Addition of map of study location will help international audience to understand the demography of the study settings and boundaries followed by the characteristics like rural and education to have a flow.
Reference for lines 89-90 should be provided.

As it is a paper of non-communicable disease what is the relevance of communicable disease like HIV prevalence twice (lines 59 and lines 92) in introduction as well as in methods?
3. Sampling:
Sampling is most likely to be multistage random sampling- district-village-household with adults being the final sampling unit. However, there is no mention about sampling and the sample size.
How households has been chosen there is no mention about it? E.g KISH method for WHO STEPS or something else.
4. No mention about confounders and effect modifiers. Comment on potential bias in the study and how recall bias was addressed?

Validity of the findings

- sample size calculation and sampling design is not mention. Comment about internal and external validity of findings in light of sample size and sampling design.

-Good explanation of absolute and relative progression rates. However,
Mention on summary of continuous variables as mean +/- SD or Median(IQR) is missing e.g., age

- Post hoc tests has already been mentioned but use of ANOVA is missing and can be
included.
- Few lines about missing data may be added for more clarity.
- Figure 1 has very nicely represented the flow. However, the number of participants in
each level is missed unlike the mention of prevalence in lines 199-200.

Additional comments

1. More emphasis has been given to statistics part of the abstract. There was no information on :
Study design, questionnaire/ study tool, inclusion and exclusion criteria in the abstract
2. General alignment of text and overall spacing must be uniform.
3. Name of the table titles needs to be improved.
E.g., Table 1: Socio-demographic and clinical characteristics of adults (N=131)
Table 2 title not appropriate – as it is about association and there is no mention about the same.
Table 3: Title not complete.
4. Figure 2 is a good thought however (prevalence and gap can be showed in the same vertical bar)
and can be compared amongst gender or age whichever is appropriate.
5. Comment on strengths also.

6. Comment about your finding in comparison with national / international perspective. Also state the
potential reasons for differences if any.
7. Few lines may be added about the generalizability of the study.
8. Funding source may be added.

·

Basic reporting

The introduction is well written but it is not clear enough to me what is the context that the authors are trying to place their work in. The paragraphs move from discussing stroke to treatment cascade and then management of hypertension, reduction of hypertension in South Africa and finally interaction of COVID-19 and hypertension. I would suggest to keep the attention of the reader in the treatment cascade for hypertension and the importance of screening, treating and controlling hypertension. I have suggested some literature to add to the references used by the authors hoping they would provide some help to the authors to place the work they have done in a better context and adding some other work done in South Africa:
- Int J Epidemiol 2014 Feb;43(1):116-28. doi: 10.1093/ije/dyt215.
- Glob Heart. 2017 Jun; 12(2): 81–90. doi:10.1016/j.gheart.2017.01.007
- BMJ Global Health 2019;4:e001386. http://dx.doi.org/10.1136/bmjgh-2018-001386
The structure of the manuscript follows a clear structure with some subtitles that help the reader to follow the text.
Tables are clear and I would like to add some minor comments that could facilitate the understanding of the content.
- Table 1 and Table 3, I would add something in the title that guides the reader about the sample used like that they are from the “Ingwavuma rural community” and where the sample comes from.
- Table 1, spell out PA
- Table 1, I would place the total in the heading of the table or the first row.
- Table 3, spell out COR and AOR maybe as a footnote
- Table 3, I would mark significant p-values in bold.
Figures are relevant, good quality and well labeled.
Raw data was supplied.

Experimental design

The work presents the issue of the treatment cascade of hypertension which is very relevant in an ageing society with increasing number of older people with hypertension and other chronic conditions.
Knowing the level of people with hypertension in rural Africa properly diagnosed, treated and controlled is crucial and this paper is covering part of the knowledge gap in this regard.
The design of the work needs to be more detailed, especially on the recruitment of participants so other researchers could replicate the work. This is explained in more detail in my comments below under General Comments.
The analysis of the study is well done, the diagnosis of the different conditions is done using the most advanced mobile equipment and the definitions of a case are following the international epidemiological standards. The study was ethically sound.

Validity of the findings

This is a cross-sectional study and the analysis performed is statistically sound.

Additional comments

Abstract
a- Last sentence of results I would rephrase the sentence to avoid using “respectively” by just moving the percentage and the AOR parenthesis just after the two variables the data refer to.
Methods
a- Line 99, at the end of the sentence there is a number that needs to be deleted. Maybe this was a reference from a previous version of the manuscript.
b- Setting is well explained and very detailed.
c- Line 106, I would suggest adding a reference for the STEPS questionnaire for those readers who are not familiar with this questionnaire.
d- Line 109, I would suggest adding “was” after “open-ended questions”.
e- Lines 114 to 116 would be easier to read if split into two sentences: one on the individuals and the other about the households and maybe starting with the households’ part.
f- Lines 116 to 119, it would be good to know when this first phase was done, whether it was done at a household visit or in the health facilities and whether it was done using the modified STEPS questionnaire. This will help to understand line 122 and 123 when authors refer to household-based surveys and measurement clinics.
Participants
a- Line 127, the authors refer to 400 participants as the population used to select the study sample but, in the abstract, they mention 393 participants. Please clarify.
b- It is not clear how these 131 adults were selected from the 393 or 400 participants and whether the level in the treatment cascade were they were found was used in the selection. For example, people who had known their hypertension condition for few years should be at the end of the treatment cascade with controlled blood pressure but those recently diagnosed possibly would only be on the screen or diagnosis phase. Please clarify.
c- If there is the need to cut words in this manuscript, this section could be reduced as most of the data is already in Table 1.
d- Line 128, I would say “Over two fifths (42.7%) were 60 plus years old, nearly a quarter (23.7%)…” I would put a full-stop after unemployment to separate socio-demographic data from measured data.
Measurements and definitions
a- The description of the different measurements is excellent.
b- I would only avoid the use of “respectively” in line 151 by placing the values after the different levels of diabetes used. The same applies to line 165 and levels of Physical activity.
c- Please add the acronym PA after Physical activity in line 160.
Outcome variables
a- Line 170, I would add “the” before “outcome”.
b- Line 173, I would rephrase the sentence to avoid using “respectively”.
Results
a- The results are well explained and the reference to tables is clearly stated.
b- Line 201, I would rephrase the sentence to avoid using “respectively”.
c- Line 213 and 215, please specify in which direction BMI is associated to awareness of hypertension. Where they “more” aware of their status?
d- Line 218, I would say “participants aged 60 years and above”
e- Under Factors associated with screening for hypertension, I would suggest presenting employment effect together and then marital status, avoiding repetitions.
Discussion and conclusions
These two sections are very well written and expressed. The authors finish the manuscript with important policy and strategic recommendations that could be useful in future primary health care planning for chronic conditions.
a- I would only recommend introducing some references on how Community Health Workers were introduced in South Africa and what are the Government expectations on their work as first line to fight NCD.
b- First and fourth paragraph (lines 238-248 and 266-274) refer to access to health care. I would recommend joining them or placing them one after the other.
c- Not sure I am following the importance of the sentence in lines 306 to 309 as an explanation of why women have higher control of their blood pressure compare to men. Moreover, the rest of the explanation makes total sense making this sentence, in my opinion, unnecessary.

---

## Round 0.2 · accepted · Accept

The reviewers and I are satisfied with the revisions the authors have implemented. The revised manuscript is now acceptable for publication. Thank you for finding PeerJ as your journal of choice. Looking forward to your future scholarly contributions.
Congratulations and very best regards.

·

Basic reporting

Well written and modified manuscript

Experimental design

Modified as suggested and satisfactory

Validity of the findings

Satisfactory modification done

Additional comments

Acceptable to the satisfaction

·

Basic reporting

The article reads well and has all the necessary elements for publication.

Experimental design

Research methods are well described and do not need further corrections.

Validity of the findings

Findings, analysis and conclusions are sound and well explained.

Additional comments

All issues raised by me in the first reading of the manuscript have been covered by the authors. No additional comments.